# Optical imaging of single-protein size, charge, mobility, and binding

Guangzhong Ma [1], Zijian Wan[1,2], Yunze Yang[1], Pengfei Zhang [1], Shaopeng Wang [1✉] &
Nongjian Tao [1,2,3]

Detection and identification of proteins are typically achieved by analyzing protein size, charge, mobility and binding to antibodies, which are critical for biomedical research and disease diagnosis and treatment. Despite the importance, measuring these quantities with one technology and at the single-molecule level has not been possible. Here we tether a protein to a surface with a flexible polymer, drive it into oscillation with an electric field, and image the oscillation with a near field optical imaging method, from which we determine the size, charge, and mobility of the protein. We also measure antibody binding and conformation changes in the protein. The work demonstrates a capability for comprehensive protein analysis and precision protein biomarker detection at the single molecule level.

[1] Biodesign Center for Biosensors and Bioelectronics, Arizona State University, Tempe, AZ 85287, USA. [2] School of Electrical, Computer and Energy Engineering, Arizona State University, Tempe, AZ 85287, USA. [3] Deceased: Nongjian Tao. ✉email: shaopeng.wang@asu.edu

Proteins play a central role in biochemical processes in living systems[1–3]. They also serve as drugs, drug targets, and disease biomarkers[4,5]. Detecting and identifying proteins are thus the most elementary tasks in biomedical research, and in disease diagnosis and therapeutics[6–8]. Various technologies have been developed for protein analysis, and the most important ones include electrophoresis, mass spectrometry (MS), enzyme-linked immunosorbent assay (ELISA) and western blot (WB)[9–13]. These technologies are often used in combination to separate and identify proteins based on charge, size (mass), and specific binding to antibodies. Although ubiquitous in both research labs and industry, they are destructive, involving protein fragmentation and denaturation[10,13]. They also lack single-molecule analysis capability required for studying heterogenous processes and for improving sensitivity in precision diagnosis[14,15].

Several technologies have been demonstrated to detect single molecules[16–28], including two label-free optical imaging methods. One is an indirect method, which takes advantage of the spectral feature of the molecules[22–24]. The other one images interference of scattered light (iSCAT), from which the protein size is quantified[20,21]. Single-molecule electrometry[29] and anti-Brownian electrokinetic (ABEL) trap[30] have been developed to measure the charge or mobility of single molecules trapped by an electric field. However, simultaneous imaging of size, charge, mobility, and binding of proteins on a single platform has not been possible. Different proteins may have similar sizes, simultaneous detection of multiple intrinsic properties of a protein is thus a key requirement for protein identification and function analysis[29,31,32]. This is an important reason that electrophoresis, MS and WB are the workhorses in biomedical research.

Here we report a method to image single proteins, measure the size and charge of each protein simultaneously, determine the mobility and analyze antibody binding of the protein in real time. Proteins in this work are resolved individually on a sensor surface, thus requiring no separation. The method is analogous to electrophoresis, MS and WB in terms of analyzing proteins based on mass (size), charge, mobility, and to ELISA in terms of antibody binding, but it is achieved with one detection platform and at the single-molecule level. We further show that the method allows label-free detection of binding-induced conformation changes in single proteins.

## Results

**Detection principles**. To achieve single-protein imaging capability, we tether proteins to an indium tin oxide (ITO) coated glass slide via polyethylene glycol (PEG), a flexible polymer, and drive the proteins into vertical oscillation by applying an alternating electric field to the ITO surface (Fig. 1a, b). To determine the oscillation, the ITO slide is placed on the objective of an inverted optical microscope, and incident light is directed onto the ITO surface via the objective to generate an evanescent field near the surface (Fig. 1a). The evanescent field is scattered by the oscillating proteins, and the scattered light is collected by the same objective, forming an image on a CMOS imager. The light intensity detected by the camera is given by (see "Methods"),

$$I = 2|u_e + u_r||u_s|\cos(\phi), \qquad (1)$$

where $u_e$, $u_s$, and $u_r$ are the evanescent wave, scattered wave, and reflected wave from ITO, respectively, and $\phi$ is the phase shift of the scattered wave relative to the combined evanescent and reflected wave. Because the evanescent field is localized near the ITO surface within ~100 nm, the scattered light is extremely sensitive to the protein-surface distance. As the protein oscillates, so does the scattered wave, which is recorded as an image sequence (Fig. 1c). We perform fast Fourier Transform (FFT) on

each pixel of the image sequence with 1 s integration time and apply a band pass filter at the frequency of the applied field to extract the oscillation amplitude while rejecting random noise (Supplementary Fig. 2). The FFT image resolves a protein as a bright spot with a parabolic tail that arises from the interference between the circular scattered wave from the protein and planar evanescent wave on the sensor surface. Spatial Fourier Transform (k-domain) of the image further reveals the characteristic two-ring feature (Fig. 1e, and "Methods" for imaging principle)[33]. The FFT image provides size, charge, and mobility of the protein as we show below.

The protein oscillation amplitude ($\Delta z_0$) is determined by the entropic force of the PEG tether and electrical driving force. The entropic force is described by the freely jointed chain (FJC) model, which predicts an entropy force proportional to the PEG displacement for small oscillation amplitude[34]. Thus, the protein oscillation amplitude is given by ("Methods")

$$\Delta z_0 = \frac{E_0(\Delta z_0)}{k_{PEG}}q, \qquad (2)$$

where $E_0(\Delta z_0)$ is the amplitude of the field, $q$ is the charge of the protein, and $k_{PEG}$ is the entropic spring constant of the PEG tether. Equation (2) shows that the oscillation amplitude is proportional to the applied electric field, but this is valid only at low fields, where the oscillation amplitude is smaller than the PEG length. When the field is sufficiently large, the oscillation amplitude reaches a plateau as the PEG tether is stretched to its most accessible length (e.g., 80%)[31,35]. This behavior has been observed for all the proteins studied here, and Fig. 1f, g shows the result for bovine serum albumin (BSA) as an example.

To quantify $\Delta z_0$ we consider that the evanescent field decays exponentially from the ITO surface into the solution with a decay length of $d$. Consequently, the FFT image contrast, $\Delta C(\Delta z_0, D_H)$, is given by (Methods),

$$\Delta C(\Delta z_0, D_H) = \beta(D_H)\left[1 - \exp\left(-\frac{\Delta z_0}{d}\right)\right], \qquad (3)$$

where $D_H$ is the protein diameter and $\beta$ is the strength of the evanescent wave scattering by a protein, which depends on the protein size. In the high-field plateau regime, the PEG tether is almost fully stretched, such that $\Delta z_0$ is close to the linear length of PEG ($L_{PEG}$), and the corresponding FFT image contrast, $\Delta C(\Delta z_0 = L_{PEG}, D_H)$, is maximum. Knowing $\Delta C(\Delta z_0 = L_{PEG}, D_H)$, Eq. (3) allows determination of $\beta$, from which $D_H$ is extracted with a calibration curve (see below).

Once $\beta$ is known, $\Delta z_0$ at different applied electric fields can be determined from the measured $C(\Delta z_0, D_H)$ with Eq. (3). This allows us to extract the charge of the protein ($q$) with Eq. (2) together with $E_0$ and $k_{PEG}$ determined with the procedures described in Methods and Supplementary Fig. 1. Finally, we obtain the protein mobility ($\mu$) from charge ($q$) and size ($D_H$) according to $\mu = q/(3\pi\eta D_H)$, where $\eta$ is the solution viscosity. We applied the method to proteins with different sizes and charges.

**Measuring size, charge, and binding of IgG**. The first example is goat immunoglobulin G (IgG), which has a molecular weight of 150 kDa and is negatively charged (pH = 7.4). Figure 2a shows the FFT image of several IgG molecules captured when the ITO surface potential is modulated with amplitude, $U_0 = 8$ V. The FFT image contrast and the extracted oscillation amplitude ($\Delta z_0$ increase with the potential amplitude and reach plateaus around 8 V (Fig. 2b). From the $\Delta z_0$ vs. $U_0$ plots we determined the size and charge of each IgG (Fig. 2c). The oscillation is in phase with the applied potential, indicating that the protein moves towards

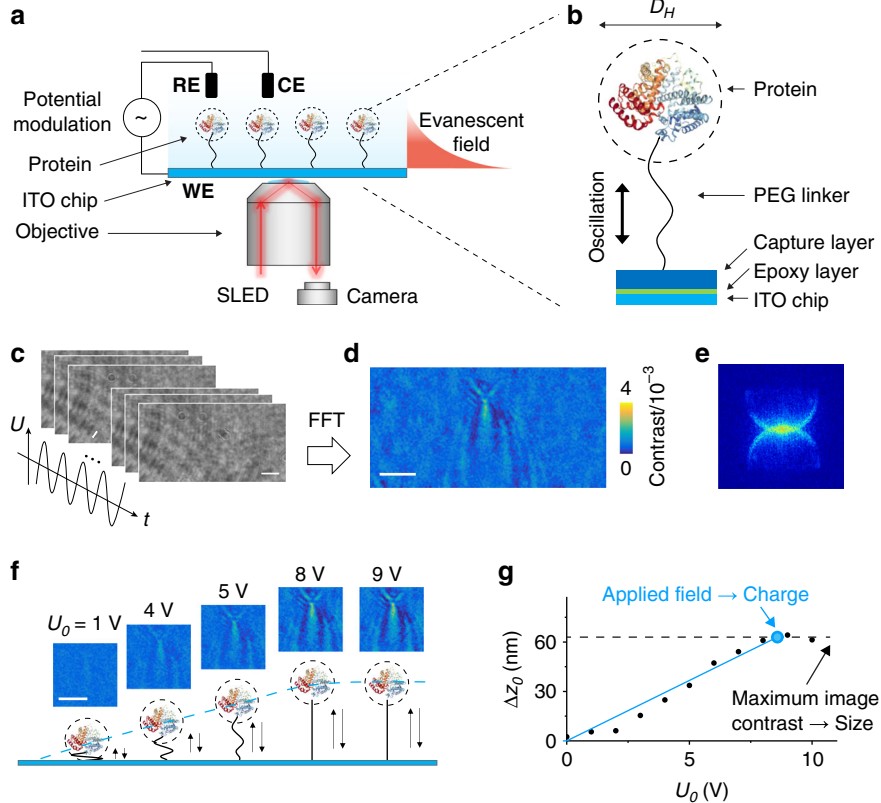

**Fig. 1 Imaging single proteins and mechanical oscillations. a** Proteins are tethered to an ITO surface with a flexible polymer tether, and driven into oscillation by an alternating electric potential applied to the surface with a three-electrode electrochemical configuration, where WE, RE, and CE are the working (the ITO surface), reference (Ag/AgCl wire) and counter electrodes (Pt coil), respectively. The oscillating protein molecules scatter the evanescent field generated by illuminating the ITO surface with light from a SLED, and the scattered light is collected to form an image captured with a CMOS camera. **b** The polymer tether is a 63 nm long polyethylene glycol (PEG), which links the protein (hydrodynamic diameter, $D_H$) to the ITO surface via surface chemistry described in "Methods". **c** Time sequence images of oscillating molecules (bovine serum albumin or BSA) recorded at 800 frames/s with potential modulation amplitude and frequency of 8 V and 80 Hz, respectively. **d** Fast Fourier Transform (FFT) filter is applied to the time sequence images shown in **c** to produce an FFT image, which resolves a single BSA molecule. **e** Spatial Fourier transform of the FFT image (k-space) in **d** showing two rings, indicating the FFT image pattern is due to the interference between a planar wave (evanescent) and circular (scattering by a molecule) waves. **f** FFT image contrast vs. potential amplitude, showing a linear regime at low electric fields, and a plateau regime associated with fully stretching of the PEG tether at high electric fields. The two regimes are indicated by the blue dash line. **g** Oscillation amplitude ($\Delta z_0$) of a BSA molecule vs. potential amplitude ($U_0$). Diameter and charge are determined from the amplitude of the plateau regime (black dash line) and the slope of the linear regime (blue line), respectively. Scale bars in **c**, **d**, **f** represent 3 µm.

the surface (thus more scattering) when the potential increases. This is expected for negatively charged IgG, which is attracted toward the surface when it is positively charged (Supplementary Fig. 2).

We analyzed 186 IgG molecules and obtained diameter, charge and mobility histograms (Fig. 2d). The histograms reveal pronounced peaks at 10.4 nm, −5.0 e (e, the elementary charge, is $1.6 \times 10^{-19}$ C) and $-0.86 \times 10^{-8}$ m$^2$ V$^{-1}$ s$^{-1}$, respectively. These mean values agree with the values measured by the dynamic light scattering experiment performed in this work (see below) and by small-angle X-ray scattering, nuclear magnetic resonance, and capillary electrophoresis reported in literature (Supplementary Tables 1, 2). The mean charge is also close to the estimated value at the buffer pH (Supplementary Table 3). The diameter histogram displays a small secondary peak located at a larger diameter, which is attributed to the formation of aggregations (Fig. 2d). Small secondary peaks also appear in the diameter and charge histograms of other proteins (e.g., Fig. 3c, Supplementary Fig. 3d, h).

To ensure imaging of single IgG molecules, we studied anti-IgG binding to the IgG tethered on the surface (Fig. 2e). This was performed by first flowing blank PBS over the surface while

driving the IgG to oscillate in the plateau regime. After reaching a stable baseline, 130 nM anti-IgG in PBS was introduced to allow binding of anti-IgG to the oscillating IgG on the surface. The FFT image contrast and the apparent diameter of IgG increase (Fig. 2f, Supplementary Fig. 4b, and Supplementary Video 1), indicating binding of anti-IgG to IgG. After measuring anti-IgG binding to IgG, blank PBS was introduced to allow study of unbinding of anti-IgG from IgG. We observed decrease in the FFT image contrast and diameter associated with the unbinding process. To ensure specific binding of anti-IgG to IgG, we performed a control experiment by introducing an antibody (antihuman IgG) that should not bind to the IgG (goat IgG). Indeed, we did not detect any change in the FFT image of the goat IgG (Supplementary Fig. 4c, d).

The standard deviations of the measured IgG diameter and charge determined from the histograms shown in Fig. 2d are 3.4 nm and 1.2 e, respectively. They reflect heterogeneity of the 186 different IgG molecules tethered on the surface, rather than measurement precision (see "Methods" and Supplementary Fig. 13 for measurement error analysis). To examine the precision in size measurements, we repeatedly measured and plotted the diameter of one IgG molecule before and after binding to an anti-

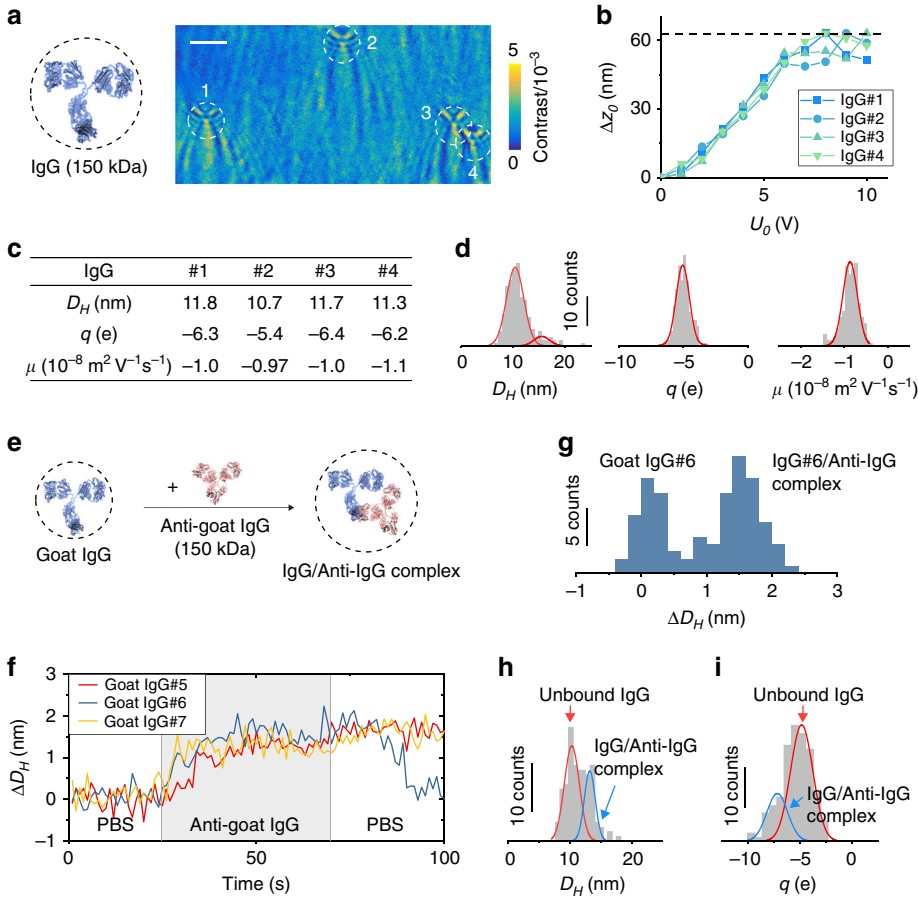

**Fig. 2 Quantifying the size, charge, and mobility of single proteins. a** FFT image of immunoglobulin G (IgG) measured with potential amplitude, $U_0 = 8$ V. Scale bar, 3 μm. **b** Oscillation amplitude ($\Delta z_0$) vs. potential amplitude ($U_0$) plots for the IgG molecules marked in **a**, from which diameter ($D_H$), charge ($q$), and mobility ($\mu$) are obtained. **c** Measured $D_H$, $q$, and $\mu$ of the molecules shown in **b**. **d** Histograms of $D_H$, $q$, and $\mu$ measured for 186 IgG molecules, where the red curves are Gaussian fittings (see Supplementary Table 4). **e** Anti-goat IgG is introduced to bind with goat IgG tethered on the surface. **f** Binding/ unbinding of anti-goat IgG with three goat IgG molecules tracked in real time, showing diameter changes associated with the binding and unbinding events. Potential amplitude: $U_0 = 9$ V. Buffer: 100× diluted PBS at pH = 7.4. Anti-goat IgG concentration:130 nM. **g** Size distribution of a goat IgG molecule (#6 shown in **f**) measured during its binding/unbinding with anti-goat IgG, where the two peaks correspond to IgG and anti-IgG/IgG complex, respectively. **h, i** $D_H$ and $q$ histograms of 137 goat IgG molecules obtained after incubation with 33 nM anti-goat IgG for ~30 min, where the two peaks are due to IgG and anti-IgG/IgG complex (see Supplementary Table 4 for the extracted mean $D_H$ and $q$; Supplementary Fig. 4a for the mobility histogram). The blue and red lines are Gaussian fittings of the peaks.

IgG, showing well separated bound and unbound states (Fig. 2g). The standard deviations of the IgG and anti-IgG/IgG molecules are 0.21 and 0.33 nm, respectively, which reflect the measurement precision. We attribute the molecular heterogeneity to the different orientations of the tethered proteins, because the NHS group in the PEG can potentially react with any amine group (lysine residue) exposed on the surface of the protein.

To further confirm the imaging of anti-IgG binding to IgG, we performed an end-point assay by incubating IgG tethered on the surface with 33 nM anti-IgG. The diameter histogram obtained after incubation shows two peaks located at 10.3 and 13.2 nm, respectively (Fig. 2h). The former is IgG, and the later corresponds to anti-IgG/IgG complex, which has an estimated mass of ~2× of IgG from the diameter. This is expected for anti-IgG/IgG complex because anti-IgG and IgG have similar masses. The charge histogram also reveals two peaks, located at −4.8 e and −7.2 e, which are associated with IgG and IgG/anti-IgG complex, respectively (Fig. 2i). As an additional control, we measured and confirmed the binding and unbinding of anti-IgG with IgG using a surface plasmon resonance setup, a well-established independent detection technology (Supplementary Fig. 5).

**Measuring size and charge of BSA and lysozyme.** We applied the method to lysozyme (MW = 14 kDa) and observed lower FFT image contrast than IgG, which is expected because of its smaller size than IgG (Supplementary Fig. 3a). The oscillation has a ~180° phase shift relative to the applied potential (Supplementary Fig. 2). This is opposite of IgG but expected because lysozyme is positively charged at pH of 7.4. Similar to IgG, the lysozyme oscillation amplitude increases with the potential modulation amplitude and then approaches a plateau at large potential amplitude (>9 V) (Supplementary Fig. 3b). We determined $D_H$, $q$, and $\mu$ of the individual lysozyme molecules and constructed histograms for these quantities (Supplementary Fig. 3c, d). The mean values of $D_H$, $q$, and $\mu$ are 4.1 nm, 4.3 e and $1.8 \times 10^{-8}$ m$^2$ V$^{-1}$ s$^{-1}$, respectively. The measured $D_H$ and $\mu$ are consistent with the dynamic light scattering data (see below) and the charge agrees with the expected value (Supplementary Table 3).

Another example is BSA (MW = 66 kDa), which is smaller than IgG but larger than lysozyme. As shown in Supplementary Fig. 3e, the BSA image contrast is lower than IgG but greater than lysozyme, which is consistent with the size of the molecule. We plotted BSA oscillation amplitude vs. potential modulation

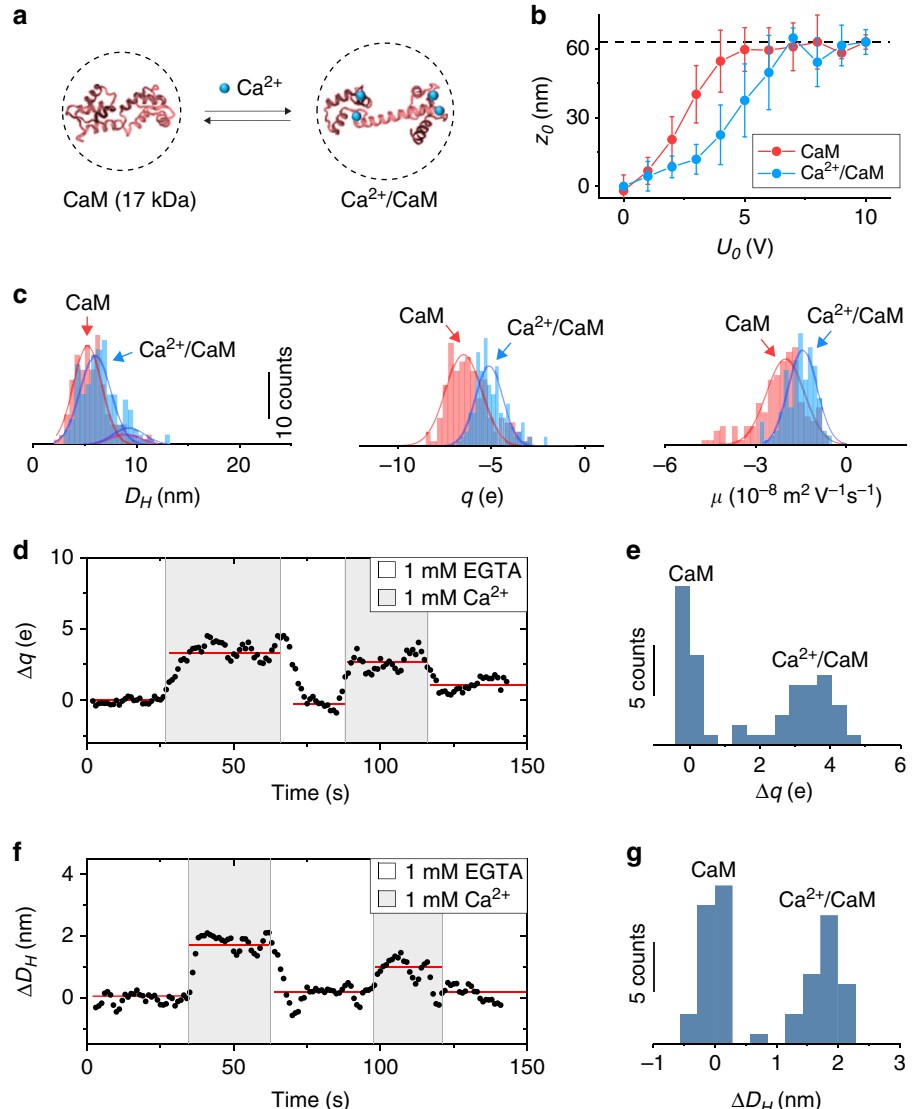

**Fig. 3 Ligand binding-induced conformation change in a protein. a** Binding of $Ca^{2+}$ to calmodulin (CaM) causes conformation and charge changes in CaM. **b** Oscillation amplitude vs. $U_0$ plots before (red) and after (blue) $Ca^{2+}$ binding to CaM, where the vertical bars are standard deviations of >150 CaM or $Ca^{2+}$/CaM molecules. **c** Statistical analysis for 150 CaM molecules (red) and 151 $Ca^{2+}$/CaM molecules (blue) showing the diameter ($D_H$), charge ($q$) and mobility ($\mu$) distributions of CaM and $Ca^{2+}$/CaM complex (see Supplementary Table 4 for a summary). The red and blue lines are Gaussian fittings of the peaks. **d, f** Tracking of the charge ($\Delta q$) and size ($\Delta D_H$) changes of a single CaM molecule induced by $Ca^{2+}$ binding over time, where the size change was measured with $U_0$ fixed at 7 V (the plateau regime) and the charge measurement was performed with $U_0 = 4$ V (the linear regime). The binding and unbinding of $Ca^{2+}$ to CaM were performed by alternatively flowing 1 mM EGTA and 1 mM $Ca^{2+}$ in 100× diluted PBS (at pH = 7.4) over the surface. The scatter plots (black dots) are raw data smoothed over three points, and the red lines are guide to the eye, showing the charge or size change duo the binding and unbinding of the molecule with $Ca^{2+}$. **e, g** Charge and size histograms obtained from the first binding and unbinding cycle in **d, f** respectively. Both histograms show two peaks corresponding to the CaM molecule with and without $Ca^{2+}$ binding.

amplitude and observed similar dependence as IgG and lysosome, namely, a low-field linear regime followed by a high-field plateau regime (Supplementary Fig. 3f). The measured $D_H$, $q$, and $\mu$ are 8.3 nm, $-5.3$ e and $-1.2 \times 10^{-8}$ m² V⁻¹ s⁻¹, respectively (Supplementary Fig. 3h). These results agree with the values from the dynamic light scattering (Fig. 4c and Supplementary Fig. 15) and calculated charge (Supplementary Table 3). We summarize the results for IgG, lysozyme and BSA, as well as other proteins and complexes in Supplementary Table 4.

**Conformation change of calmodulin**. In addition to the size, charge and mobility of single proteins, the present method provides valuable information on understanding the conformation changes in proteins. For most proteins, the conformation change is

expected to be accompanied by size and/or charge changes and can thus be determined by the present method. To demonstrate this capability, we studied $Ca^{2+}$ binding to calmodulin (CaM), a protein that mediates various $Ca^{2+}$ signaling processes, such as muscle contraction, inflammation, and fertilization[36]. CaM has two globular domains, each containing two EF-hand motifs, so it can bind up to four $Ca^{2+}$ and causes conformation and charge changes in CaM (Fig. 3a)[37]. We tethered CaM to an ITO surface, incubated it in buffers with and without $Ca^{2+}$, and measured the oscillation vs. potential modulation amplitude in each buffer (Fig. 3b). CaM with $Ca^{2+}$ reaches a plateau at a larger potential amplitude than that without $Ca^{2+}$. This is because $Ca^{2+}$ binding reduces the net charge of CaM. We determined $D_H$, $q$, and $\mu$ for CaM and $Ca^{2+}$/ CaM complex from the oscillation vs. potential modulation

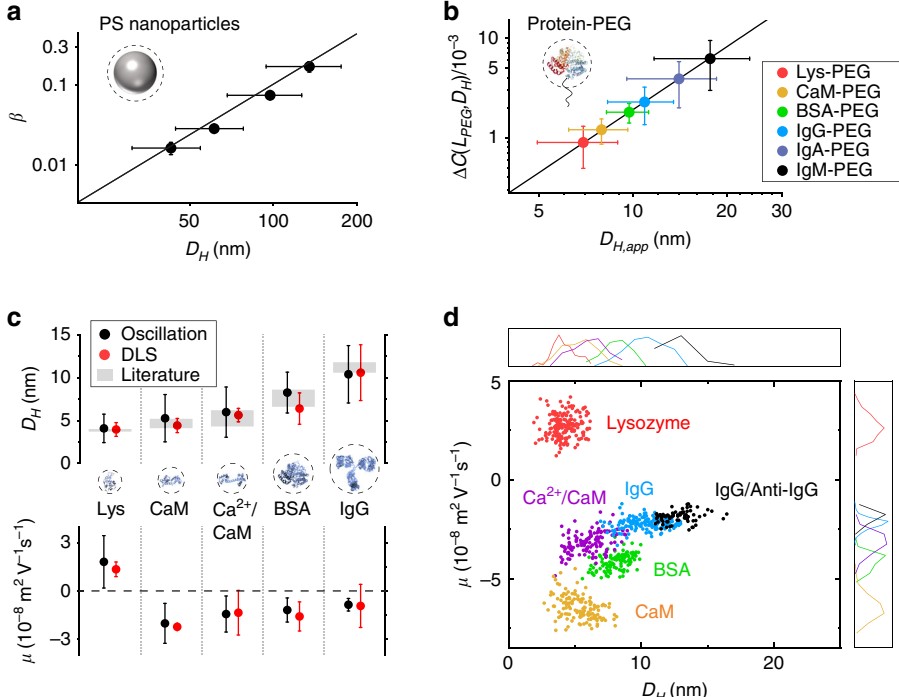

**Fig. 4 Identifying proteins based on size and mobility. a** Image contrast vs. size for polystyrene (PS) particles. The solid line is fitting of the data showing a power relation of 2.1. Because PS particles bind to the ITO surface from the bulk solution ($\Delta z_0 \rightarrow \infty$), the image contrast is $\beta$ according to Eq. (3). **b** Determining protein–PEG complex size ($D_{H,app}$) from FFT image contrast change, $\Delta C(L_{PEG}, D_H)$. Unlike PS particles, proteins are tethered to the surface and driven into oscillation with a maximum oscillation amplitude determined by the PEG length. **c** Comparison of measured $D_H$ and $\mu$ (Oscillation) with those measured by dynamic light scattering (DLS) experiments performed here and reported in literature. **d** Mobility ($\mu$) − size ($D_H$) plot of single proteins and protein–ligand complexes, showing different proteins or complexes are separated in the 2D-plot, which resembles "2-D electrophoresis". Mobility or size, each alone, cannot clearly separate the different proteins, as shown by the corresponding 1D size and mobility histograms (top and side panels). The error bars for image contrast, size and mobility represent standard deviations of >100 single-nanoparticle or single-protein measurements.

amplitude plot and obtained histograms from hundreds of CaM and Ca$^{2+}$/CaM complex images (Fig. 3c). The results show that $D_H$ of CaM increases from 5.3 to 6.0 nm upon binding to Ca$^{2+}$, which is consistent with literature values[37] and attributed to Ca$^{2+}$ binding-induced conformation change in CaM. $q$ and $\mu$ for CaM are $-6.5$ e and $-2.0 \times 10^{-8}$ m$^2$ V$^{-1}$ s$^{-1}$, respectively, which change to $-5.1$e and $-1.4 \times 10^{-8}$ m$^2$V$^{-1}$ s$^{-1}$ after binding to Ca$^{2+}$. We further verified the size and charge changes by performing dynamic light scattering measurement (Fig. 4c).

To demonstrate the capability of real-time conformation and charge detection, we also monitored Ca$^{2+}$ binding to CaM by first driving CaM into oscillation, and then alternatively flowing 1 mM Ca$^{2+}$ and 1 mM ethylene glycol tetraacetic acid (EGTA) solutions over the surface. EGTA is known to cause unbinding of Ca$^{2+}$ from CaM via chelation with Ca$^{2+}$, so the experiment allowed us to repeatedly monitor the binding and unbinding of Ca$^{2+}$ to a CaM molecule and the associated size and charge changes of CaM (Fig. 3d–g, Supplementary Fig. 6, and Supplementary Video 2). The real-time data are consistent with the end-point measurements carried out by incubating CaM in Ca$^{2+}$ and Ca$^{2+}$ free solutions. We determined the measurement precision by measuring the size and charge of the same single molecule repeatedly. The statistics (Fig. 3e, g) shows that the standard deviations for the size and charge are 0.3 nm and 0.5 e, respectively. The fundamental limit of this technology is shot noise (Methods). As such, a better a signal-to-noise ratio (SNR) is expected by increasing the incident light intensity. This will allow us to detect smaller molecules and improve the precision by minimizing the effect of ITO charging (Methods).

**Size calibration**. The protein size ($D_H$) in this work is determined with calibration by imaging polystyrene (PS) nanoparticles of different diameters ($D_H = 40$–140 nm). These nanoparticles are larger than the proteins and can be directly imaged with the setup by subtracting the background from each image, allowing us to obtain the image contrasts vs. size (Fig. 4a). We confirmed that the image contrasts were due to single particles rather than aggregates with scanning electron microscopy (SEM) (Supplementary Fig. 7). The power relation between the image contrast and $D_H$ is ~2.1 ± 0.2, smaller than 3 expected from a simple scattering model. This discrepancy is attributed to the roughness of the ITO surface as shown by the AFM experiment and simulation (Supplementary Fig. 8). Electric double layer-enhanced light scattering has been reported, which may also contribute to the observed optical signals here[27]. The calibration curve was used to determine the size of protein–PEG complex ($D_{H,app}$) (Fig. 4b). After subtracting the size of PEG (Methods), the size of protein was obtained as shown in Supplementary Table 4. To narrow size gap between the proteins and nanoparticles, we measured large macromolecules, immunoglobulin A (IgA) and immunoglobulin M (IgM), which are dimer and pentamer of IgG, respectively. The results are well predicted by the calibration curve (Fig. 4b and Supplementary Fig. 9). The diameter and mobility measured for single proteins in each case agree with those by reference technologies in literature (Supplementary Tables 1, 2), providing validation of the calibration curve.

**2D identification of single proteins**. Two-dimensional (2D) gel electrophoresis is a powerful technology that identifies proteins

based on size and isoelectric point (or mobility at different pH). The present single-molecule imaging method can perform protein analysis in an analogous manner at the single-molecule level. This capability is shown in Fig. 4d, which plots different proteins and protein–ligand complexes according to mobility and size. The proteins and complexes in the 2D-plot are well separated, allowing identification of proteins like 2D electrophoresis. Binding of IgG to anti-IgG shifts the IgG region to a new position in the 2D-plot, which is analogous to the WB technology, providing further identification of the protein. Figure 4d also shows that 1D histograms based on either size or mobility alone are not capable to differentiate different proteins and complexes, which underscores the importance of simultaneous size, charge and mobility measurements.

Covalently tethering the proteins to the surface allows us to continuously track the same molecules for sufficiently long time and switch solutions in the measurements without losing the molecules, which is a challenging task for most single-molecule trapping methods. Compared to single-molecule methods that do not require surface attachment such as ABEL trap and iSCAT, the tethered proteins are more likely to have fixed orientations, thus the broad distribution in measured size and charge reflect the orientational heterogeneity of the proteins that otherwise hard to obtain. For example, lysozyme has a more asymmetrical shape than calmodulin, and it shows a smaller standard deviation in size. Since the PEG tethers the protein via nonselective NHS-amine coupling, it is possible the PEG alters the conformation of the protein or blocks the binding sites to ligands. Another limitation of our technology is low time resolution (1 s), which prevents recording of fast biological processes.

We have developed an optical imaging technology to quantify the size, mobility, and charge of single proteins simultaneously, and to monitor protein–ligand interactions and associated conformation changes in proteins. These capabilities are analogous to the widely used electrophoresis, ELISA, and WB technologies, but achieved with one detection platform and at the single-molecule level. We anticipate that our technology will complement the traditional technologies and enable the study of various processes of proteins, especially low volume samples (e.g., single cells and exosomes) including conformation changes, molecular binding and posttranslational modifications of proteins, and to detect drug–target interactions and disease biomarkers at the single-molecule level.

## Methods

**Materials.** ITO slides with resistance of 70–100 $\Omega$ were purchased from SPI Supplies. Streptavidin was purchased from VWR. (3-Glycidyloxypropyl)tri-methoxylsilane, lysozyme, calmodulin, and BSA were purchased from Sigma-Aldrich. Goat IgG (anti-digoxigenin) was purchased from Abcam. Goat antihuman IgG and rabbit anti-goat IgG were purchased from Invitrogen. Secretory IgA (from human colostrum, MW = 385 kDa) and IgM (from human plasma, MW = 950 kDa) were purchased from Athens Research and Technology. Polystyrene nanoparticles were purchased from Bangs Labs and the hydrodynamic diameters were determined with dynamic light scattering. Biotin–PEG–NHS (MW = 10 kDa) was purchased from Nanocs. Deionized (DI) water with resistivity of 18.2 M$\Omega$ cm was used in all the experiments.

**Experimental setup.** The imaging setup was built on an inverted microscope (Olympus IX-81) with a 60× (NA = 1.49) oil immersion objective. A super-luminescent light emitting diode (SLED) (SLD-260-HP-TOW-PD-670, Superlum) with central wavelength at 670 nm and output power of up to 15 mW was used as light source. A CMOS camera (ORCA-Flash 4.0, Hamamatsu) was used to record 2048 by 256 pixels images at 800 frames/s. A sinusoidal potential ($f$ = 80 Hz) was applied to the ITO slide with a function generator (33521A, Agilent) and a potentiostat (AFCBP1, Pine Instrument Company) using a three-electrode configuration, where the ITO, a Ag/AgCl wire and Pt coil served as the working, reference and counter electrodes, respectively. A USB data acquisition card (NI USB-6251, National Instruments) was used to synchronize the applied potential, the current, and the recorded images.

**Modification of ITO surface.** ITO slides were cleaned by sonication sequentially in acetone, ethanol, and DI water, each for 20 min, and then soaked in $H_2O_2$/ $NH_3 \cdot H_2O/H_2O$ (1:3:5) for 1 h, which were then rinsed with DI water and dried with $N_2$. The slides were incubated in isopropanol containing 1% (3-Glycidyloxypropyl)trimethoxylilane for 10 h to silanize and form terminal epoxy groups on the surface. The epoxy-functionalized ITO slides were rinsed with isopropanol and DI water, dried with $N_2$, and incubated in 0.1 mg/ml streptavidin + 1× PBS for 2 h. At last, the ITO slides were incubated in 0.1 mg/ml BSA + 1× PBS for 20 min.

**Tethering of proteins to ITO surface.** Biotin–PEG–NHS (~15% length variation) was used to tether the protein to the streptavidin-functionalized ITO surface. The protein (IgG, lysozyme, BSA, CaM, IgA, or IgM) was first incubated with the biotin–PEG–NHS tether at 10:1 ratio to form a PEG–protein complex in 1× PBS overnight at 4 °C. The solution containing protein–PEG complex was then added to streptavidin coated ITO slides and incubated for 2 h to allow biotin–streptavidin binding. Finally, the ITO slide was rinsed with 100× diluted PBS to remove free protein molecules in the solution.

**Calibration curve.** 100× diluted PBS was placed on top of the ITO surface, and PS nanoparticle solution was added to allow binding of the nanoparticles to the surface. An image sequence was recorded at 800 frames/s for 5 s. The hydrodynamic diameter of each PS nanoparticle sample was measured with dynamic light scattering.

**Signal processing.** An 80 Hz FFT filter (bandwidth = 1 Hz) was applied to the recorded image sequence in time domain to remove random noises (Supplementary Fig. 2) using MATLAB. The temporal FFT image was converted to k-space using ImageJ, and a spatial filter was applied to remove spatial noises (Supplementary Fig. 10i). A region of interest (ROI) with 10 × 10 pixels was selected for each protein, and the mean intensity within the ROI ($I_p$) was used to determine the contrast of the protein. An adjacent region of the same size was selected as a reference region, and the mean intensity of the reference region ($I_r$) was also determined. The contrast of the protein was determined with $\Delta C(\Delta z_0, D_H) = (I_p - I_r)/I$, where $I$ is the mean intensity within the ROI without FFT filter. The size and charge of each protein were determined based on the FFT image contrast.

**Equation of motion of tethered protein molecules.** The motion of a protein tethered to a surface by PEG is determined by,

$$m\frac{d^2z}{dt^2} + c\frac{dz}{dt} + k_{PEG}z = qE + F_r, \qquad (4)$$

where $m$, $z$, $c$, and $q$ are the mass, displacement, damping coefficient, and charge of the protein, $k_{PEG}$ is the spring constant of the PEG tether, arising from the entropic force, $E$ is the applied electric field. $F_r$ is the stochastic force on the PEG and the protein due to thermal fluctuation, which has an average of zero. For a protein with molecular weight of 100 kDa ($m = 1.7 \times 10^{-19}$ g) under an electric field oscillating at 80 Hz, the first and the second terms are ~$10^{-1}$ and $10^{-3}$ pN, respectively, which are much smaller than the entropic force estimated below.

To estimate the entropic force associated with the conformation change of the PEG tether, we use the FJC model[34,38], which leads to an entropic force of

$$f_{entropy} = k_{PEG}z = \frac{3k_BT}{nb^2}z, \qquad (5)$$

where $k_B$ is the Boltzmann constant, $T$ is temperature, $b$ is the Kuhn length of PEG, and $n$ is the number of segments with length of $b$. For PEG10k, $b = 0.55$ nm, $n = 113$[39], and $k_{PEG} = 3.62 \times 10^{-4}$ N/m. The entropic force is 22.8 pN when the PEG is stretched to 63 nm, which is many orders of magnitude greater than the first and second terms of Eq. (4). This simplifies Eq. (4) to $k_{PEG}z = qE$, which is Eq. (2). For a sinusoidal electrical field, $E = E_0e^{j\omega t}$, where $E_0$ is the field amplitude and $\omega = 2\pi f$ is the angular frequency. The sinusoidal oscillation of the displacement with amplitude $\Delta z_0$ takes the form of $z = \Delta z_0 e^{j(\omega t + \theta)}$, where $\theta$ is the phase shift of the oscillation with respect to the field. Both the oscillation amplitude and phase shift can be determined by performing FFT on the image sequence.

**Measuring the applied electric field and protein charge.** Because the ionic concentration varies with the distance from the surface, the applied electric field varies with the molecule-surface distance, which is denoted as $E = E_0(\Delta z_0, U_0)e^{j\omega t}$. To measure the electric field, we tethered 40 nm streptavidin coated gold nanoparticles (AuNPs) to a surface with PEG10k linkers[31], swept the potential negatively from 0 to −2 V, and recorded particle-surface distance ($\Delta z_0$) change vs. potential ($U_0$) (Supplementary Fig. 1a, b). Because AuNPs are negatively charged at pH = 7.4, the negative potential sweep pushes them away from the surface, leading to decrease in the image intensity (Supplementary Fig. 1c). The intensity decreases exponentially with the potential and reaches a minimum when the PEG is fully stretched (Supplementary Fig. 1d). By converting the image intensity into particle-surface distance ($\Delta z_0$)[31], we obtained $\Delta z_0$ vs. $U_0$ for the AuNP (Supplementary Fig. 1e). The plot shows a linear regime followed by a plateau regime, which is observed for all the proteins studied here (Figs. 2b, 3b,

Supplementary Fig. 3b, f). At the transition from the linear to plateau regimes, the polymer is almost fully stretched ($\Delta z_0 = L_{PEG}$) and the entropic force reaches a maximum. According to Eq. (2), by knowing the entropic force at $\Delta z_0 = L_{PEG}$ and charge of the AuNP, the electric field at distance $\Delta z_0 = L_{PEG}$ can be determined, which is denoted as $E_0(\Delta z_0 = L_{PEG}, U_0 = U_{trans})$, where $U_{trans}$ is the potential amplitude at the transition point (Supplementary Fig. 1e). The charge ($q_{NP}$) of the AuNPs is related to the zeta potential ($\zeta$) according to[40,41],

$$q_{NP} = 4\pi a^2 \cdot \frac{2\varepsilon_r \varepsilon_0 \kappa k_B T}{Ze} \sinh\left(\frac{Ze\zeta}{2k_B T}\right)\left[1 + \frac{1}{\kappa a \cdot \cosh^2(Ze\zeta/4k_B T)}\right], \quad (6)$$

where $a$ is the radius of the particle, $Z$ is the valence of ions in the solution, $\varepsilon_0$ and $\varepsilon_r$ are the permittivity of vacuum and the relative permittivity of the solution, $\kappa^{-1}$ is the Debye length and $e$ is the elementary electric charge. $\kappa^{-1} = 7.89$ nm (see "charge screening effect" section) and $\zeta = -13.1 \pm 0.7$ mV, as measured by electrophoretic light scattering, from which we obtained $q_{NP} = -42.5 \pm 2.3$ e with Eq. (6). The entropic force of the stretched PEG is 22.8 pN. Knowing the charge and entropic force, we found $E_0(\Delta z_0 = L_{PEG}, U_0 = -0.95 \text{ V}) = -3.35 \times 10^6$ V/m. Because the applied electric field scales with potential, we have

$$E_0(\Delta z_0 = L_{PEG}, U_0 = U_{trans}) = 3.53 \times 10^6 \, U_{trans}/\text{m}. \quad (7)$$

**Effect of PEG tether**. The size obtained here includes contribution from the PEG tether. To extract the diameter of the protein ($D_H$), the following equation is used,

$$D_H^3 = D_{H,app}^3 - D_{H,PEG}^3, \quad (8)$$

where $D_{H,app}$ is the apparent diameter of the protein–PEG complex, and $D_{H,PEG}$ is the diameter of PEG coil measured with dynamic light scattering.

**Length of PEG tether**. The PEG tether used in this work has a molecular weight of 10 kDa, which consists of 225 ethylene glycol units, each has a length of 0.278 nm[42,43]. The linear length of the PEG is thus 0.278 nm × 225 = 63 nm.

**Extracting oscillation amplitude with FFT**. The image sequence records the oscillation of protein molecules over time. Plotting the local image contrast vs. time reveals periodic oscillation of a protein (e.g., IgG and lysozyme) (red dashed line, Supplementary Fig. 2a, c). The FFT amplitude spectrum shows a sharp peak located at the frequency of the applied electric field (red line, Supplementary Fig. 2b, d). We performed this FFT analysis on each pixel of the image sequence (Fig. 1c), extracted the oscillation amplitude averaged over 1 s, and constructed an FFT image (oscillation amplitude image) shown in Fig. 1d.

**Probability of anti-IgG binding to IgG**. We estimated probability ($P(t)$) of anti-IgG binding to IgG by measuring the kinetic constants with surface plasmon resonance (Supplementary Fig. 5). In terms of the kinetic constants, $P(t)$ takes the form of,

$$P(t) = \frac{k_a[\text{anti-IgG}]}{[\text{anti-IgG}]k_a + k_d}\left[1 - e^{-([\text{anti-IgG}]k_a + k_d)t}\right], \quad (9)$$

where [anti-IgG] is the concentration of anti-IgG in the solution, and $k_a$ and $k_d$ in Eq. (9) are the association and dissociation rate constants, respectively. In the anti-IgG binding experiment shown in Fig. 2e, [anti-IgG] = 130 nM and $t \sim 50$ s, which leads to $P(t) = 0.36$ according Eq. (9), indicating 36% chance of an IgG binding with an anti-IgG.

**Additional data of anti-IgG binding to IgG**. Using the measured size and charge (Fig. 2h–i), we determined the mobility for IgG and IgG/anti-IgG complex and the mobility histogram (Supplementary Fig. 4a). Unlike the size and charge histograms, the mobility shows only one peak. This is because mobility is an intensive quantity, which scales with $q/D_H$. Supplementary Fig. 4b shows snapshots of the IgG molecules in Fig. 2f captured before (10 s), during (60 s), and after (90 s) the binding experiment. A control experiment performed by flowing 33 nM antihuman IgG over the goat IgG shows no detectable changes in the image intensities (Supplementary Fig. 4c, d).

**Confirmation of single nanoparticles with SEM**. We obtained contrast images of PS nanoparticles of different sizes and imaged the same ROI with SEM to confirm that each pattern in the contrast images was due to single particle rather than aggregates. The contrast images show good correlation with the SEM images as marked by the red arrows in Supplementary Fig. 7a–d. We selected single particles, measured the sizes, and plotted the size vs. contrast in logarithmic scale (Supplementary Fig. 7e). The result shows that the slope is ~2, which is consistent with what we observed in Fig. 4a.

**Imaging principle**. Light is directed on the ITO surface at an angle close to the critical angle to generate an evanescent wave ($u_e$) propagating along the surface. When a protein is present on the surface, it scatters the evanescent wave,

generating a scattered wave ($u_s$), given by

$$u_s(r, r') = \beta u_e(r')e^{-d|r-r'|}e^{-ik|r-r'|}, \quad (10)$$

where $r'$ is the location of the protein and $r$ is the location on the image, $\beta$ is a scattering coefficient related to the polarizability of the molecule, $d$ is the decaying constant of the evanescent wave, and $k$ is the wave number of evanescent wave. The superposition of the two waves, together with light reflected from the ITO surface ($u_r$), is

$$u(r, r') = u_r(r) + u_e(r) + u_s(r, r'). \quad (11)$$

The overall reflected light detected by the camera ($I$) is given by[44–52],

$$I = |u_r(r) + u_e(r) + u_s(r, r')|^2. \quad (12)$$

The image contrast of the particle is described by

$$I(r, r') = |u_r(r) + u_e(r) + u_s(r, r')|^2 - |u_r(r) + u_e(r)|^2, \quad (13)$$

where the last term is the background image in the absence of the protein. For weak scattering, $|u_s|^2$ is small, and Eq. (13) is reduced to

$$I(r, r') = 2|u_r(r) + u_e(r)||u_s(r, r')|\cos(\varphi), \quad (14)$$

which shows that the image contrast is originated from the interference between $u_r + u_e$ and $u_s$ (where $\varphi$ is the phase shift). Using Eq. (14), we computed an image, which closely resembles the experimental image (Supplementary Fig. 8b). Equation (14) shows that the image intensity scales with $\beta$, which is proportional to the cubic power of the diameter ($D_H^3$). The observed size dependence of the image contrast is slower than cubic power (between 2 and 3), which is attributed to surface roughness, as discussed below.

**Effect of surface roughness**. The above analysis assumes a perfect surface. In practice, ITO surface has finite roughness and atomic force microscopy (AFM) reveals grain-like features on the surface (Supplementary Fig. 8a). The surface roughness is particularly important for small objects, such as protein molecules, which are comparable with or smaller than the grains. We simulate the surface roughness effect by including an additional term, $u_{rough}$ in Eq. (13),

$$I(r, r') = \left|u_r(r) + u_e(r) + u_{rough}(r, r') + u_s(r, r')\right|^2$$
$$- \left|u_r(r) + u_e(r) + u_{rough}(r, r')\right|^2, \quad (15)$$

which leads to increased background and also slower dependence of the image contrast on the protein size. Using the grain size of the ITO measured from the AFM images, we performed numerical simulation of the size dependence of the image contrast. The simulation used 1000 small polystyrene particles randomly distributed on the surface around a polystyrene particle of interest with diameter varying from 20 to 150 nm. The size distribution of the small particles used for simulation was based on the AFM measurement, which varied from 0 to 25 nm, with an average diameter of 5 nm. The logarithmic plot of the image contrast vs. diameter shows a slope of ~2.2 (Supplementary Fig. 8c).

**Measuring the size of IgA and IgM**. To further confirm that the image contrasts are due to single molecules, we measured the image contrasts of immunoglobulin A (IgA) and immunoglobulin M (IgM) (Supplementary Fig. 9), which are dimer and pentamer of IgG, and determined sizes using the image contrast vs. size plot (Fig. 4b). The size of IgA and IgM are determined to be 13.9 ± 4.5 and 17.6 ± 6.0 nm, respectively, close to the values measured with dynamic light scattering (14.3 ± 4.5 and 21.4 ± 6.0 nm).

**Surface charging effect and background noise**. A bare ITO surface also responds to the applied electric field and gives rise to a background. This response arises from the charge-dependent refractive index of ITO. To evaluate this effect, we modulated a bare ITO slide with potential ($U_0 = 10$ V and $f = 80$ Hz) and obtained the FFT images (Supplementary Fig. 10a). The features shown in the images are due to the grains of the ITO surface, which affect limits of detection for the diameter ($D_H$) and charge ($q$). We converted the features in the ITO background image to the equivalent $D_H$ and $q$ noise images using the calibration curve in Fig. 4b and the Einstein equation (assume the mobility is $1 \times 10^{-8}$ m$^2$ V$^{-1}$ s$^{-1}$, the typical value for proteins[49]). The results show the distributions of $D_H$ and $q$ associated with surface roughness (Supplementary Fig. 10b, c).

In single-molecule detection, the charge-induced features overlap with the protein images and affect detection accuracy (Supplementary Fig. 10g). To reduce this effect, we performed $k$-space FFT to the temporal FFT image and converted the image to $k$-space, which shows two rings originated from the interference of scattered field and evanescent field (Supplementary Fig. 10h)[33]. We apply a filter to block part of the low frequency region (Supplementary Fig. 10i), where the background features are located, and the result shows most of the features are removed (Supplementary Fig. 10j). We applied the same filter to the images in Supplementary Fig. 10a–c to remove the background features and obtained the distributions of $D_H$ and $q$, from which the limits of detection were estimated to be ~1.0 nm for diameter, and 0.3 e for charge (Supplementary Fig. 10d–f).

**SNR estimation**. The evanescent wave ($E_e$) propagates along the ITO surface and is scattered by the sample particle or protein on the surface. The scattered wave ($E_s$) is given by[50,51]

$$E_s = \frac{k^3}{4\pi} \alpha E_e \frac{e^{-dr_{xy}}}{kr_{xy}} e^{-i(kr_{xy} + \delta)}, \quad (16)$$

$$E_e = E_e^o e^{-iky}, \quad (17)$$

where $k = 2\pi/\lambda$ is the wave number, $\alpha$ is the polarizability of the particle, $r_{xy}$ is the distance from the particle to the point of the field on the surface, $d$ is the decay constant of evanescent field, and $\delta$ is phase shift between the plane wave and the circular wave (scattered wave). The polarizability $\alpha$ is given by

$$\alpha = 4\pi a^3 \frac{n_P^2 - n_{H_2O}^2}{n_P^2 + 2n_{H_2O}^2}, \quad (18)$$

where $a$ is the radius of the particle, $n_p$ and $n_{H_2O}$ are the refractive index of particle and water, respectively. For a polystyrene (PS) particle with radius of 6 nm, $\alpha$ is about $4 \times 10^{-25}$ m$^3$.

The overall light collected by the camera ($I_{total}(\theta, \varphi)$) is the superposition of $E_e$, $E_s$, and light reflected by the ITO surface ($E_r$),

$$I_{total}(\theta, \varphi) \propto |E_e + E_r + E_s|^2 = |E_e + E_r|^2 + 2\text{Re}\left\{E_s\left(E_e^* + E_r^*\right)\right\} + |E_s^2|. \quad (19)$$

The first term ($|E_e + E_r|^2$) in Eq. (19) is the background without the particle, and the third term is negligible for small particles. We note that all the particles in this work are sufficiently small. Equation (19) can thus be simplified as

$$I(\theta, \varphi) \propto 2\text{Re}\left\{E_s\left(E_e^* + E_r^*\right)\right\}. \quad (20)$$

By combining with Eqs. (16) and (17), Eq. (20) can be written as

$$I(\theta, \varphi) \propto 2 \cdot \frac{k^3}{4\pi} \alpha E_e^o (E_e^o + E_r^o) \frac{e^{-dr_{xy}}}{kr_{xy}} \cos[k(y + r_{xy}) + \delta]. \quad (21)$$

The evanescent light intensity has ~5 times of enhancement over the incident light ($I_i$), so $E_e^o \sim \sqrt{5}E_i^o$. The incident light is fixed at an angle where the intensity of reflection is a half of that of incident light, which leads to $E_r^o = \frac{1}{\sqrt{2}}E_i^o \sim \frac{1}{\sqrt{10}}E_e^o$. The maximum value of $I(\theta, \varphi)$ is obtained at $e^{-dr_{xy}} = 1$ and $\cos[k(y + r_{xy}) + \delta] = 1$,

$$I_{max}(\theta, \varphi) = 2\frac{k^3}{4\pi} \alpha E_e^o (1 + \frac{1}{\sqrt{10}}) E_e^o \frac{1}{kr_{xy}} = \frac{6.5k^2\alpha}{2\pi r_{xy}} I_i. \quad (22)$$

In terms of number of photons, the above equation can be written as

$$N_{max} = \frac{6.5k^2\alpha}{2\pi r_{xy}} N_i. \quad (23)$$

$r_{xy}$ is about a half of the ROI (the particle is in the center of ROI), which is ~0.25 μm From the incident light intensity, $N_i$ is determined to be ~$2 \times 10^{15}$ photons/s for an illumination area of ~300 μm × 300 μm and. The number of photons on the ROI (0.5 μm × 0.5 μm), $N_{i,ROI}$, is $5 \times 10^9$ photons/s. The number of scattered photons for the particle, according to Eq. (23), is $7 \times 10^5$ photons/s. The shot noise is $\sim\sqrt{0.5N_{i,ROI}} = 5 \times 10^4$, and the corresponding SNR is ~14. We also determined the noise using our experimental data, which showed shot noise limitation (Supplementary Fig. 14).

Based on the above analysis, the scattering cross section for a 6 nm PS particle is $\frac{6.5k^2\alpha}{2\pi r_{xy}} \times r_{xy}^2 = 3.5 \times 10^{-5}$ μm$^2$, much greater than that estimated by Piliarik and Sandoghdar in iSCAT[20]. One reason for this difference is that evanescent wave scattering can greatly enhance the scattering cross section. It has been reported that over three orders of magnitude scattering cross section enhancement for nanoparticles in evanescent field generated by p-polarized excitation[52,53]. It is possible that other less understood effects also contribute to the scattering cross section enhancement given that the surface condition is more complex than that in iSCAT. The enhanced cross section allows more photons to be scattered and interfere with the evanescent field, which provides sufficient signal for single-protein detection.

**Contrast estimation**. We calculate contrast ($C$) with the FFT images, which is defined by

$$C = \frac{|E_e + E_r + E_s|^2 - |E_e + E_r|^2}{|E_e + E_r|^2} \sim \frac{2\text{Re}\left\{E_s(E_e^* + E_r^*)\right\}}{|E_e + E_r|^2}. \quad (24)$$

$|E_e + E_r|^2$ is about a half of the intensity of incident light $\frac{1}{2}|E_i|^2$. According to Eqs. (16)–(23), Eq. (24) leads to

$$C_{max} = \frac{2\frac{k^3}{4\pi}\alpha E_e^o(1 + \frac{1}{\sqrt{10}})E_e^o\frac{1}{kr_{xy}}}{\frac{1}{2}|E_i|^2} = \frac{6.5k^2\alpha}{\pi r_{xy}} = 3 \times 10^{-4}. \quad (25)$$

In reality, the contrast is proportional to $(ka)^{2.1}$ rather than $(ka)^3$, thus $C_{max}$ is ~$4 \times 10^{-3}$.

**Charge screening effect**. Effective charges are measured here, which are related to the net charges by,

$$\frac{\sigma_{eff}}{\sigma_{total}} = \frac{\zeta}{\psi} = e^{-\kappa x}, \quad (26)$$

where $\sigma_{eff}/\sigma_{total}$ is the ratio of the effective charge density to the net charge density, $\zeta$ is zeta potential of the protein, $\psi$ is the potential at the protein surface, $x$ is the slipping layer thickness, and $\kappa^{-1}$ is the Debye length, which is determined by the Debye–Hückel equation,

$$\kappa^{-1} = \sqrt{\frac{\varepsilon\varepsilon_0 k_B T}{2N_A e^2 I}}, \quad (27)$$

where $\varepsilon$ is the dielectric constant of the buffer, $\varepsilon_0$ is the permittivity of free space, $k_B$ is the Boltzmann constant, $N_A$ is the Avogadro number, $e$ is the elementary charge, and $I$ is the ionic strength. Using Eq. (26), the slipping layer thickness of protein molecules can be determined by the zeta potential and Debye length at two different concentrations with,

$$x = \frac{\ln(\zeta_1/\zeta_2)}{\kappa_2 - \kappa_1}. \quad (28)$$

We measured the zeta potential of lysozyme in 1× PBS and 100× diluted PBS, which are $\zeta_1 = 2.78$ mV and $\zeta_2 = 13.8$ mV, respectively. The corresponding Debye lengths are $\kappa_1^{-1} = 0.789$ nm and $\kappa_2^{-1} = 7.89$ nm according to Eq. (27), from which the slipping layer thickness for the protein is determined to be 1.4 nm. Using these parameters, we plotted $\sigma_{eff}/\sigma_{total}$ ratio vs. ionic strength (Supplementary Fig. 11), showing that the effective charge is ~20% of total charge for 1× PBS, and ~90% in 100× diluted PBS.

**Measurement error**. The derivation of size and charge involves several parameters of the PEG and the applied field, thus the variation of these parameters can affect the accuracy in size and charge. We estimate the variations here. (a) PEG length. The PEG10k has a length variation of ~15%, which leads to ~10% variation in the measured intensity according to Eq. (3). Because intensity is proportional to the square of diameter, the variation in determined diameter is ~3.3%. The PEG length is also related to the spring constant, which could affect the interpretation of charge (see below). (b) PEG spring constant. Each PEG tether may have different spring constant due to the variation in length and different local environment. The variation in spring constant is ~7% measured by AFM force spectroscopy[54]. (c) Determining the plateau of oscillation. The plateau and the transition point in the oscillation amplitude vs. applied potential plot (Fig. 1g) are used to determine the size and charge. The error in determining the maximum extension is ~10%, which leads to 3.3% variation in size and 10% variation in the potential (or charge). (d) Measuring the applied electric field. The field is determined based on the charge of gold nanoparticles, which has a variation of 5%. (e) Surface charging effect introduces 1 nm error in size and 0.3 e error in charge. Based on the above analysis, the cumulative error for size and charge are determined to be $0.05D_H + 1$ nm and $0.13|q| + 0.3$ e, respectively. The cumulative error for size and charge are small compared to the standard deviation of the measured values, which suggests our measurements resolve molecular heterogeneity. Details of the calculation are shown in Supplementary Note 1, and the results are summarized in Supplementary Fig. 13.

## Data availability

The data that support the findings of this study are available from the corresponding author upon reasonable request.

## Code availability

MATLAB code for fast Fourier Transform is provided in Supplementary Note 2.

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

## Acknowledgements

This paper is dedicated to the memory of N.T. (1963–2020). The present work was financially supported by Gordon and Betty Moore Foundation, and National Institute of Health (R44GM126720). We thank Dr. Dong-Kyun Seo and Shaojiang Chen at Arizona State University for their help with dynamic light scattering measurement, and Dr. Shuoxing Jiang and Xu Zhou for help with AFM.

## Author contributions

G.M. carried out the experiments and analyzed the data, Z.W. helped some experiments, Y.Y. and P.Z. helped data analysis, S.W. helped with instrumentation, N.T. and S.W. conceived and supervised the project, and G.M., S.W., and N.T. wrote the paper.

## Competing interests

A US patent application (16/584,120) has been filed by Arizona Board of Regents on behalf of Arizona State University based on an early draft of this article and published on 03/26/2020. Inventors: N.T. and G.M.
