## [Peer Review File · Nature Communications]

REVIEWERS' COMMENTS:

Reviewer #2 (Remarks to the Author):

The authors have satisfactorily addressed nearly all of my comments. This is a great story and should be published in Nature Comm as soon as possible, but there is one lingering issue:

In response to my original comment “Page 6, Line 18: I think the claim that the observed heterogeneity is coming from molecule heterogeneity (as opposed to measurement precision) is not justified. The repeated measurements from the same molecule are an important step, but do not conclusively show this. Different molecules may also have different tether lengths (the PEG is polydisperse), different protein binding positions (the NHS is not specific), different ITO surface environments (the authors showed this in Sup Fig 8). What about the accuracy of determining when maximum extension has been reached since there is considerably noise in the plot (Fig 2b). All of these effects would have to be shown to be negligible (through either calculation or experiment) before this claim is justified.”,

the authors have eloquently responded to my concerns, and the new Figure S13 clearly shows that some heterogeneity remains after accounting for measurement precision. However, the text in the response (that addresses each of the 4 points), should also be added to the SI, not just the figure. The authors should simply adapt their response text to text for the SI. I do not need to review this again, they just not to move this text into the SI so that the reader can understand the rationale (which is convincing).

Reviewer #3 (Remarks to the Author):

The authors have done a commendable job addressing the numerous comments from previous referees. While they may seem excessive, they are clearly important in the context of the claims made in the original manuscript.

There is one aspect that I do believe still ought to be addressed: The authors maintain that the origin of their optical contrast is single molecule scattering. This continues to puzzle me for two reasons: (a) their calculation requires a scattering cross section increase by more than 6 (!) orders of magnitude, which seems very, very large indeed, even in the context of the cited paper. (b) I don't understand why their signal scales with the square of protein size rather than the volume, if indeed we are seeing single molecule scattering. It appears much, much more likely that what the authors are seeing is a secondary effect, such as displacement of surface ions or similar. It would be beneficial to discuss this issue explicitly in the text - or at the very least explicitly state that their data only makes sense in the context of a 10^{-5} um^2 cross section, so that the general reader is aware.

RESPONSE TO REVIEWERS' COMMENTS:

Reviewer #2 (Remarks to the Author):

The authors have satisfactorily addressed nearly all of my comments. This is a great story and should be published in Nature Comm as soon as possible, but there is one lingering issue:

In response to my original comment “Page 6, Line 18: I think the claim that the observed heterogeneity is coming from molecule heterogeneity (as opposed to measurement precision) is not justified. The repeated measurements from the same molecule are an important step, but do not conclusively show this. Different molecules may also have different tether lengths (the PEG is polydisperse), different protein binding positions (the NHS is not specific), different ITO surface environments (the authors showed this in Sup Fig 8). What about the accuracy of determining when maximum extension has been reached since there is considerably noise in the plot (Fig 2b). All of these effects would have to be shown to be negligible (through either calculation or experiment) before this claim is justified.”, the authors have eloquently responded to my concerns, and the new Figure S13 clearly shows that some heterogeneity remains after accounting for measurement precision. However, the text in the response (that addresses each of the 4 points), should also be added to the SI, not just the figure. The authors should simply adapt their response text to text for the SI. I do not need to review this again, they just not to move this text into the SI so that the reader can understand the rationale (which is convincing).

Response: We have included the response text in Supplementary Note 1.

Reviewer #3 (Remarks to the Author):

The authors have done a commendable job addressing the numerous comments from previous referees. While they may seem excessive, they are clearly important in the context of the claims made in the original manuscript.

There is one aspect that I do believe still ought to be addressed: The authors maintain that the original of their optical contrast is single molecule scattering. This continues to puzzle me for two reasons: (a) their calculation requires a scattering cross section increase by more than 6 (!) orders of magnitude, which seems very, very large indeed, even in the context of the cited paper. (b) I don't understand why their signal scales with the square of protein size rather than the volume, if indeed we are seeing single molecule scattering. It appears much, much more likely that what the authors are seeing is a secondary effect, such as displacement of surface ions or similar. It would be beneficial to discuss this issue explicitly in the text - or at the very least explicitly state that their data only makes sense in the context of a 10^{-5} um^2 cross section, so that the general reader is aware.

Response: (a) We agree with the reviewer that it is beneficial to point out this issue in the paper. We have clarified that there could be other less understood reasons for the strong cross section enhancement, and only with the enhanced cross section we can image single molecules (page 24, following the calculations). (b) Our SEM results (Fig. S7) confirm the signal scales with the

square of the particle size rather than the volume. We attribute this observation to surface roughness effect, which is supported by our AFM measurement and simulation (Fig. S8).